# Disodium Fumarate Alleviates Endoplasmic Reticulum Stress, Mitochondrial Damage, and Oxidative Stress Induced by the High-Concentrate Diet in the Mammary Gland Tissue of Hu Sheep

**DOI:** 10.3390/antiox12020223

**Published:** 2023-01-18

**Authors:** Meijuan Meng, Xu Zhao, Ran Huo, Xuerui Li, Guangjun Chang, Xiangzhen Shen

**Affiliations:** Ministry of Education Joint International Research Laboratory of Animal Health and Food Safety, College of Veterinary Medicine, Nanjing Agricultural University, Nanjing 210095, China

**Keywords:** high-concentrate diet, disodium fumarate, Ca^2+^, mitochondria, endoplasmic reticulum, oxidative stress

## Abstract

The long-term feeding of the high-concentrate diet (HC) reduced rumen pH and induced subacute rumen acidosis (SARA), leading to mammary gland tissue damage among ruminants. Disodium fumarate enhanced rumen bufferation and alleviated a decrease in rumen pH induced by the HC diet. Therefore, the purpose of this study was to investigate whether disodium fumarate could alleviate endoplasmic reticulum (ER) stress, mitochondrial damage, and oxidative stress induced by the high-concentrate diet in the mammary gland tissue of Hu sheep. In this study, 18 Hu sheep in mid-lactation were randomly divided into three groups: one fed with a low-concentrate diet (LC) diet, one fed with a HC diet, and one fed with a HC diet with disodium fumarate (AHC). Each sheep was given an additional 10 g of disodium fumarate/day. The experiment lasted for eight weeks. After the experiment, rumen fluid, blood, and mammary gland tissue were collected. The results show that, compared with the LC diet, the HC diet could reduce rumen pH, and the pH below 5.6 was more than 3 h, and the LPS content of blood and rumen fluid in HC the diet was significantly higher than in the LC diet. This indicates that the HC diet induced SARA in Hu sheep. However, the supplementation of disodium fumarate in the HC diet increased the rumen pH and decreased the content of LPS in blood and rumen fluid. Compared with the LC diet, the HC diet increased Ca^2+^ content in mammary gland tissue. However, the AHC diet decreased Ca^2+^ content. The HC diet induced ER stress in mammary gland tissue by increasing the mRNA and protein expressions of GRP78, CHOP, PERK, ATF6, and IRE1α. The HC diet also activated the IP3R-VDAC1-MCU channel and lead to mitochondrial damage by inhibiting mitochondrial fusion and promoting mitochondrial division, while disodium fumarate could alleviate these changes. In addition, disodium fumarate alleviated oxidative stress induced by the HC diet by activating Nrf2 signaling and reducing ROS production in mammary gland tissue. In conclusion, the supplementation of disodium fumarate at a daily dose of 10 g/sheep enhanced rumen bufferation by maintaining the ruminal pH above 6 and reduced LPS concentration in ruminal fluid and blood. This reaction avoided the negative effect observed by non-supplemented sheep that were fed with a high-concentrate diet involving endoplasmic reticulum stress, oxidative stress, and mitochondrial damage in the mammary gland tissue of Hu sheep.

## 1. Introduction

The roughage is the main body of the ruminant diet, and concentrate feed is only a beneficial supplement for high yield performance. Due to the influence of factors such as geography and climate, high-quality herbage resources are lacking in our country. Ruminants are usually fed large amounts of the high-concentrate diet to improve performance. The long-term feeding of diets with a high-carbohydrate ratio can accelerate the fermentation rate of microorganisms in the rumen, which leads to the accumulation of VFA in the rumen, a decrease in pH, and finally causes subacute rumen acidosis (SARA). SARA is characterized by daily episodes (>2 h) of low ruminal pH in the range of 5.2 and 6, and it is difficult to diagnose because of the absence of overt clinical signs, also making it very difficult to detect in a short period of time [1,2]. It has seriously affected the health and production of ruminants, which has become one of the bottlenecks seriously restricting the development of animal husbandry [3]. Therefore, improving the performance of animals while maintaining their health under the high-concentrate diet feeding mode has become an urgent problem which needs to be studied and solved in the current breeding process.

When SARA occurs, the homeostasis of mammary epithelial cells is broken. Ca^2+^ disorder can cause a series of health problems [4,5], such as inflammation, oxidative damage, mitochondrial dysfunction, and so on. The endoplasmic reticulum (ER) and mitochondria are the storage sites of Ca^2+^ in cells. ER stress can lead to the release of Ca^2+^ in the ER, and the continuous release of Ca^2+^ from ER may initiate the absorption of Ca^2+^ by mitochondria. When an increase in the concentration of Ca^2+^ in the cytoplasm exceeds the buffer capacity of mitochondria, cell damage can be induced. The mitochondria-associated endoplasmic reticulum membrane (MAM) is located between mitochondria and the endoplasmic reticulum, which makes ER-mitochondrial Ca^2+^ transport possible [6]. Mitochondria represent the main structures regulating calcium homeostasis in cells, and cellular calcium overload can lead to oxidative stress, resulting in the production of a large number of reactive oxygen species (ROS) [7,8]. The overproduction of ROS can destroy the inherent antioxidant defense system of cells and cause oxidative stress. Excess ROS can further cause mitochondrial damage, which leads to their abnormal fusion and fission [9]. Excessive mitochondrial division can induce cell apoptosis [10,11]. Our previous studies show that high-concentrate diets increased the content of Ca^2+^ in the mammary gland tissue of dairy cows, and induced oxidative stress, ER stress, and cell apoptosis in mammary gland tissues [12,13]. During in vitro experiments, LPS increased Ca^2+^ levels in bovine mammary epithelial cells, resulting in intracellular Ca^2+^ disorder and inducing cell damage [14]. Therefore, the destruction of calcium homeostasis is the key link of mammary gland injury, and the protection of mammary gland tissue from oxidative stress and the maintenance of normal functions of mitochondria and endoplasmic reticulum are critical in the prevention of mastitis.

Fumaric acid has powerful free-radical-scavenging properties, as well as various anti-inflammatory, antioxidant, and immunomodulatory effects [15,16,17]. Fumaric acid can prevent cadmium-induced liver damage by reducing oxidative stress [18] and can play a protective role in the brain by activating the Nrf2 signaling pathway [19]. Fumaric acid can not only improve rumen fermentation, but also improve the utilization ability of lactic acid utilization bacteria in the rumen, thus reducing the deposition of lactic acid in the body [20,21]. However, adding fumaric acid to ruminant diets can significantly decrease the pH value in the fermentation system. Therefore, fumaric acid is detrimental to ruminal fermentation in ruminants, which are dominated by high-concentrate diets. Fumaric acid is detrimental to ruminal fermentation in ruminants, which are dominated by high concentrates. Disodium fumarate not only has the effect of fumaric acid, but also is a strong base and weak acid, which has a certain buffering effect [22]. Research has indicated that disodium fumarate has positive effects on rumen fermentation. Zhou et al. showed that disodium fumarate supplementation at a level of 20 g/d increases the ruminal fluid pH from 6.74 to 6.94 and alters microbial populations in Hu sheep fed with a high-forage diet [23]. Dietary disodium fumarate can promote cellulose fermentation and enrich microbial species, which is beneficial to improve rumen pH under the high-concentrate conditions [24,25,26]. Yu et al. also showed that supplementing the diet with fumarate (6 g/head per day) had significant effects on rumen microbial fermentation by decreasing ammonia and branched-chain VFA, by increasing acetate and propionate, and by increasing NDF digestion [25]. At present, there are relatively few studies which focus on the effect of disodium fumarate on the damage of mammary tissue induced by the high-concentrate diet in Hu sheep. Therefore, we investigated whether the supplementation of disodium fumarate in a high-concentrate diet could alleviate mammary gland injury in Hu sheep fed with a high-concentrate diet.

We hypothesized that the supplementation (10 g/day) of disodium fumarate in Hu sheep feeding with a high-concentrate diet prevents the drop in rumen pH to risk levels for SARA presentation, avoiding endoplasmic reticulum stress, oxidative stress, and mitochondrial damage in mammary gland tissue.

## 2. Materials and Methods

### 2.1. Ethics Statement

The study was conducted following the guidelines of the Experimental Animals of the Ministry of Science and Technology pertaining to the care and use of animals for experimental and scientific purposes (2006, Beijing, China). Nanjing Agricultural University’s Research Ethics Committee of Animal Care and Use approved the study (NAJU.PZ2020102). Our experiment lasted for 56 days, from 29 December 2021 to 22 February 2022, at the Hu sheep breeding base (Huzhou, Zhejiang, China).

### 2.2. Animals and Diets

In total, 18 lactation Hu sheep (2-3 weeks post-partum, 0.558 ± 0.2 kg/d of milk yield, and 1.47 ± 0.15 kg of DMI), with an average live weight of 49.80 ± 5.00 kg, were randomly selected and randomly divided into 3 groups (n = 6 per group). All the sheep were fitted with a rumen fistula before 3 weeks of the formal trial. The operation was strictly directed by the standard surgical procedures and performed by professional technicians in accordance with animal welfare procedures. After surgery, rectal temperatures of all animals were monitored daily to ensure that all sheep were healthy for the official trial (39.0 ± 0.2 °C). All the sheep did not show any clinical signs of infection throughout the trial period. The sheep were kept under identical conditions and individually fed in an indoor pen (0.97 m × 2.82 m).

They were fed 3 experimental diets. The concentrate (19.28% maize, 4.20% soybean meal, 2.70% wheat bran, and 1.20% rapeseed meal) and forage (35% peanut vine, and 35% cron stover silage) were combined to obtain a low-concentrate diet (30:70). The concentrate (46.86% maize, 9.80% soybean meal, 8.10% wheat bran, and 2.8% rapeseed meal) and forage (15%, peanut vine, 15% cron stover silage) were combined to obtain a high concentrate diet (70:30). The composition and nutritive value of this diet are shown in Table 1. The treatments were as follows: (1) a low-concentrate (LC) diet; (2) a high-concentrate (HC) diet; and (3) a high-concentrate (AHC) diet supplemented with disodium fumarate at daily dose of 10 g sodium fumarate/sheep. The disodium fumarate was added by mixing with the concentrate before each feeding, and it was purchased from Shanghai Macklin Biochemical Co., Ltd. (Shanghai, China).

In order to avoid experimental errors and ensure that all sheep achieve a similar metabolic state before the formal experiment, all sheep were fed the LC diet for 3 weeks as the adaptation period. Three experiment diets (LC, HC and AHC) were then fed for a formal trial period of 8 weeks. All the sheep were provided with clean freshwater ad libitum and fed daily at 08:00 and 16.00. They were milked daily at 18:00 in their own indoor pen. To avoid confounding variables, such as daily management and handling, all the sheep were fed and milked by experienced farm staff. In order to ensure that the experimental sheep could eat enough diets, they were given at 110% of the previous day’s intake. The DMI was recorded daily and is listed in Table 2. All sheep were healthy, and their body condition was monitored daily by evaluating rectal temperature and feed intake.

### 2.3. Sample Collection and Treatment

The rumen fluid of all sheep was sampled through the rumen fistula at 1, 2, 3, 4, 5, 6, and 8 h after feeding (from 9:00 to 16:00) on the on the last three days of week 8 (54, 55 and 56 day of the trail). Moreover, 0 h was indicated in the sampling period. Rumen fluid was collected at the same time on all three days. Rumen fluid was filtered with four layers of gauze. The rumen pH was immediately measured using the pH meter (HI9125, HANNA instruments, Cluj-Napoca, Romania).

On the morning of day 57, the blood was collected from the jugular vein of all sheep before slaughter on an empty stomach. All sheep were fasted at 21:00 on day 56, and blood samples were collected at 9:00 on day 57. The blood samples were centrifuged at 3000 r/min for 15 min, and the plasma was collected and stored at −20 °C.

The LPS concentrations in the plasma and rumen fluid were measured by the instructions of the chromogenic endpoint tachypleus amebocyte lysate assay kit (Xiamen Limulus Reagent Experimental Factory Co., Ltd. Xiamen, China). More details of the process are reported in a previous report [9].

Prior to slaughter, the sheep were fasted for 12 h with access to clean freshwater and the standard procedures to slaughter were followed [27]. At 09:00 on day 57, firstly, each sheep was anesthetized by an intramuscular injection of 0.002 mL/kg of xylazine hydrochloride (Jilin Huamu Animal Health Products Co., Ltd. Jilin, China). Then, they were slaughtered by exsanguination, as performed by professional technicians under the supervision of veterinarians in accordance with animal welfare procedures. The mammary gland samples were collected from the left udder of parenchyma (same position) in all the sheep after phosphate-buffered saline washing. Portions of the mammary gland tissue were fixed with 4% paraformaldehyde solution for histological analysis (approximately 1 cm). Other samples were taken and placed in a −80 °C freezer for subsequent mRNA, protein, and antioxidant index (SOD, MDA and GSH) assays.

### 2.4. The Determination of CAT, GSH, and SOD

The reduced glutathione (GSH) and superoxide dismutase (SOD) activities and malondialdehyde (MDA) content were analyzed in the blood and mammary gland tissue using corresponding commercial kits (SOD, S0101S; MDA, S0131S, Beyotime Biotechnology institute, Shanghai, China; GSH, A006-2-1; Nanjing Jiancheng Bioengineering Institute, Nanjing, China) according to the manufacturer’s instruction.

### 2.5. RNA Analysis

After the breast tissue of Hu sheep was removed from the −80 freezer, about 50 mg of the sample was weighed and 1 mL of Trizol was added to extract the total RNA. After the concentration of total RNA was detected by Nano Drop 2000 (Thermo Scientific, Waltham, MA, USA), the RNA was reversely transcribed into cDNA using the reverse transcription reagent Hifair^®^ II First-Strand cDNA Synthesis SuperMix for qPCR according to the instructions (11120ES60, Yeasen Biotechnology (Shanghai) Co., Ltd. Shanghai, China). Finally, the ChamQ Universal SYBR qPCR Master Mix Kit was used for real-time qPCR on ABI 7300 PCR instrument (Q711, Vazyme, Nanjing, China). We selected the glyceraldehyde phosphate dehydrogenase (GAPDH) as the internal reference gene, and the relative quantitative method 2^−ΔΔCt^ was used for quantitative analysis. The primers used in the experiment were designed online on NCBI and synthesized by Shanghai Gerui Bioengineering Co., Ltd (Shanghai, China). The primer sequence is shown in Appendix A.

### 2.6. Western Blot

The mammary gland tissue was removed from the −80 °C freezer and weighed. Afterwards, 1 mL of RIPA lysate and 10 μL of PMSF were added to every 100 mg of mammary gland and homogenized in a homogenator until the liquid became transparent. Then, the lysate was transferred to a 1.5 mL centrifuge tube and centrifuged at 12,000 rpm/min at 4 °C for 15 min. After centrifugation, the supernatant was collected as the protein sample. The protein concentration was determined by the BCA Kit (20201ES76, Yeasen Biotechnology (Shanghai) Co., Ltd. Shanghai, China), and the protein concentration was diluted to 2 μg/μL. Then, 10 μL of the prepared protein samples was electrophoretic on 10% SDS-PAGE. The protein was transferred to the PVDF membrane via the sandwich method in the transmembrane apparatus. After the transfer, the PVDF membrane was sealed in 75% milk at room temperature for 2 h, which was then incubated in the corresponding antibody at 4 °C overnight. On the second day, the PVDF membrane was removed from the primary antibody and incubated in the corresponding secondary antibody at room temperature for 2 h in a shaker. After incubation, it was washed with TBST again. The protein bands were added to the ECL chemiluminescence solution, and the ChemiDoc MP system was used (Bio-rad, Berkeley, CA, USA) to visualize the abundance signals. The gray values of each strip were analyzed and quantified by Image lab software. The antibodies used in the experiment are shown in Appendix A.

### 2.7. Immunohistochemical Analysis

The collected mammary gland tissue was immersed in paraformaldehyde for 24 h. The sample was first dehydrated with anhydrous ethanol, then embedded in paraffin and cut into 5 mm slices with a cutting machine. The paraffin sections were sliced, and sections were blocked for 30 min using blocking solution. Primary antibodies were added and incubated overnight at 4 °C for 24 h, and then the mixture was incubated for 1 h at room temperature in the dark after the addition of secondary antibody. Subsequently, the sections were washed with PBS. The sections were counterstained with hematoxylin for 3 min, dehydrated, and covered with a coverslip. A final drop of sealed tablets was added, followed by optical microscopy.

### 2.8. Immunofluorescence Analysis

The prepared paraffin sections were permeated with 0.5% Triton X-100 at room temperature for 10 min. The sections were then treated with citrate buffer for antigenic repair, followed by incubation with 3% H_2_O_2_ for 30 min and closed with 1% BSA for 30 min at room temperature. The sections were incubated with the corresponding primary antibody at 4 °C overnight. On the second day, the sections were washed with PBS and incubated with the corresponding fluorescent secondary antibody. After cleaning with PBS, the nuclei were stained with a DAPI solution under the condition of avoiding light. After cleaning again with PBS, anti-fluorescence quenching was added to seal the tablet. The fluorescent images were visualized by a LSM 710 confocal laser microscope system (Zeiss, Oberkochen, Germany). All antibodies used in this part were listed in Appendix A.

### 2.9. Ca^2+^ in Mammary Gland Tissue

The calcium content in the mammary gland tissue of Hu sheep was determined by using the calcium colorimetric assay kit (S1063S, Beyotime Biotechnology, Shanghai, China). The specific steps are as follows. The mammary gland tissue was cut into pieces. Then, 20 mg of tissue was added with a 100 μL sample cracking solution and homogenized by homogenizer until fully cracked. Then, the sample was centrifuged at 4 °C 10,000× *g* for 5 min, and the supernatant was taken and placed on the ice for testing. Then, the 50 μL standard or sample was added to each well of the 96-well plate, and the 150 μL test solution was then added to each well and mixed well. The sample was incubated at room temperature without light for 5~10 min. The absorbance at 575 nm was measured with an enzyme marker to make a standard curve. Finally, the calcium content in mammary gland tissue was calculated according to the formula given in the instructions.

### 2.10. Statistical Analysis

A general linear model was applied with repeated measures to analyze the rumen pH data by IBM SPSS 19.0 Statistics for Windows (IBM Inc., New York, NY, USA), in which the effects of Hu sheep and that of diet and time were considered random and fixed effects, respectively. The interaction between diet and time was also analyzed. The sheep and measured time were considered to be repeated measurements. The coefficient correlations were analyzed using bivariate correlations in SPSS 19.0 Statistics. The data on the expression of mRNA and protein, the concentration of Ca^2+^ and antioxidant capacity, were analyzed using the individual sheep as the experimental unit. These statistical data were analyzed via the independent-samples *t*-test in SPSS 19.0 Statistics for Windows. All data are expressed as the mean ± SEM. *p* < 0.05 indicates a significant difference.

## 3. Results

### 3.1. Supplemental Disodium Fumarate Effects on Ruminal pH, Ruminal Fluid, and Blood LPS Concentration, Calcium Content in Mammary Gland Tissue and Dry Matter Intake

Compared with the LC diet, the HC diet tended to reduce the DMI of Hu sheep. However, there was no significant difference in DMIs among the three diets (*p* > 0.05). Compared with the LC diet, the HC diet could reduce the rumen pH, and the rumen pH was below 5.6 for more than 3 h after 0~8 h feeding, indicating that we successfully induced subacute rumen acidosis (SARA) in Hu sheep. However, the rumen pH of the AHC diet was higher than that of the HC diet after the addition of disodium fumarate, and both were higher than 5.6 (Figure 1A). The concentration of LPS in blood and rumen fluid were significantly elevated in the HC group compared to the LC group (*p* < 0.01). However, compared with the HC group, the AHC group significantly decrease the LPS concentration in blood and rumen fluid (Figure 1B,C, *p* < 0.05 or *p* < 0.01). Compared with the LC diet, the HC diet significantly increased the Ca^2+^ concentration in the mammary gland tissue of Hu sheep (*p* < 0.01). However, the AHC diet significantly decreased the Ca^2+^ content of mammary tissue in Hu sheep (Figure 1D, *p* < 0.01).

### 3.2. Supplemental Disodium Fumarate Effects on Endoplasmic Reticulum Stress Indicators in Mammary Gland Tissue

As shown in Figure 2, compared with the LC diet, the HC diet significantly increased the mRNA expression of PERK (*p* < 0.01), IRE1α (*p* < 0.01), ATF6 (*p* < 0.01), ATF4 (*p* < 0.01), eIF-2α (*p* < 0.01), GRP78 (*p* < 0.05), and CHOP (*p* < 0.05). However, the supplementation of disodium fumarate in the HC diet significantly reduced the mRNA expression of PERK (*p* < 0.05), IRE1α (*p* < 0.01), ATF6 (*p* < 0.01), ATF4 (*p* < 0.01), GRP78 (*p* < 0.05), and CHOP (Figure 2A,B, *p* < 0.01). Western blot results show that the HC diet significantly reduced the protein expression of p-PERK (*p* < 0.01), PERK (*p* < 0.01), p-IRE1α (*p* < 0.01), IRE1α (*p* < 0.05), ATF6 (*p* < 0.01), ATF4 (*p* < 0.01), GRP78 (*p* < 0.01), eIF-2α (*p* < 0.05), and CHOP (*p* < 0.05), and the AHC diet significantly decreased p-PERK (*p* < 0.05), PERK (*p* < 0.01), p-IRE1α (*p* < 0.01), IRE1α (*p* < 0.05), ATF6 (*p* < 0.01), ATF4 (*p* < 0.01), GRP78 (*p* < 0.01), eIF-2α (*p* < 0.01), and CHOP (Figure 2C–M, *p* < 0.05). In addition, the immunohistochemical results further indicate that, compared with the HC diet, the AHC diet could reduce the protein expression of GRP78 in the mammary gland tissue of Hu sheep (Figure 2N). In conclusion, the high-concentrate diet induced ER stress in the mammary gland tissue of Hu sheep, and the supplementation of disodium fumarate in the HC diet alleviated ER stress in the mammary gland tissue of Hu sheep.

### 3.3. Supplemental Disodium Fumarate Effects on IP3R-VDAC1-MCU Expression in Mammary Gland Tissue

Compared with the LC diet, the HC diet significantly increased the protein expression of GRP75, VDAC1, and IP3R (*p* < 0.01), and the protein expressions of GRP75, VDAC1, and IP3R were significantly decreased by adding disodium fumarate into the HC diet (Figure 3A–D, *p* < 0.01 or *p* < 0.05). The results of immunofluorescence further show that, compared with the LC diet, the HC diet increased the fluorescence intensity of IP3R, GRP75, and VDAC1, as well as the interactions between IP3R and VDAC1, and between GRP75 and VDAC1. However, this phenomenon can be reversed by adding disodium fumarate to the HC diet (Figure 3E,F). In addition, compared with the LC diet, the HC diet significantly increased the protein expression of MCU (*p* < 0.01), while the AHC diet significantly decreased the protein expression of MCU (Figure 3G,H, *p* < 0.01). Immunofluorescence results further verified the results of Western blot. Compared with the LC diet, the HC diet increased the protein expression of MCU, which could be reversed by supplementing disodium fumarate in the HC diet (Figure 3I).

### 3.4. Supplemental Disodium Fumarate Effects on Mitochondrial Biogenesis in Mammary Gland Tissue

Compared with the LC diet, the HC diet significantly reduced SIRT1 (*p* < 0.01), PGC-1α (*p* < 0.01), NRF1 (*p* < 0.01), and TFAM (*p* < 0.05). However, the AHC diet significantly increased the protein expression of SIRT1 (*p* < 0.05), PGC-1α (*p* < 0.01), NRF1 (*p* < 0.01), and TFAM (Figure 4A–E, *p* < 0.05). The immunohistochemical results further indicated that, compared with the LC diet, the HC diet could reduce the protein expression of SIRT1 and PGC-1α in the mammary gland tissue of Hu sheep, while the AHC diet could reverse this phenomenon (Figure 4F).

### 3.5. Supplemental Disodium Fumarate Effects on Mitochondrial Dynamics in Mammary Gland

Compared with the LC diet, the HC diet significantly decreased the protein expression of mitochondrial fusion proteins OPA1, MFN1, and MFN2 (*p* < 0.01), and the protein expressions of OPA1, MFN1, and MFN2 could be significantly up-regulated in the AHC diet (Figure 5A–D, *p* < 0.01). Compared with the LC diet, the HC diet significantly increased the protein expressions of Drp1, Fis1, and MFF (*p* < 0.01 or *p* < 0.05), and the protein expressions of Drp1, Fis1, and MFF were significantly decreased in the AHC diet (Figure 5A,E–G, *p* < 0.01 or *p* < 0.05). Immunohistochemical results further indicate that, compared with the LC diet, the HC diet could decrease the protein expression of MFN2 and increase the protein expression of Drp1, while adding disodium fumarate in the HC diet could increase the protein expression of MFN2 and decrease the protein expression of Drp1 (Figure 5H,I). These results indicate that the HC diet promoted mitochondrial division and inhibited mitochondrial fusion, thus causing mitochondrial damage, and disodium fumarate supplementation in the HC diet reversed this phenomenon.

### 3.6. Supplemental Disodium Fumarate Effects on Oxidative Stress of Mammary Gland Tissue

Compared with the LC diet, the HC diet significantly decreased the mRNA expression of GSS, CAT, GPX1, AKT1, HO-1, and NQO1 (*p* < 0.01 or *p* < 0.05). However, the supplementation of disodium fumarate in the HC diet significantly increased the mRNA expressions of GSS, CAT, GPX1, AKT1, HO-1, and NQO1 (Figure 6A, *p* < 0.01 or *p* < 0.05). Compared with the LC diet, the HC diet significantly decreased the activity of SOD and increased the content of MDA in blood and mammary gland tissue, and the HC diet also significantly decreased the content of GSH in blood (*p* < 0.01). However, the supplementation of disodium fumarate in the HC diet reversed the occurrence of this phenomenon. The AHC diet significantly increased the activity of SOD and decreased the content of MDA in the blood and mammary gland tissue of Hu sheep (*p* < 0.01 or *p* < 0.05) and increased the GSH content of blood in Hu sheep (Figure 6B–G, *p* < 0.01).

Western blot results show that, compared with the LC diet, the HC diet significantly decreased the protein expressions of Nrf2, p-Nrf2, GPX1, CAT, NQO1, and HO-1, while AHC significantly increased the protein expressions of Nrf2, p-Nrf2, GPX1, CAT, NQO1, and HO-1 (Figure 7A–G, *p* < 0.01 or *p* < 0.05). In addition, the immunohistochemical results show that, compared with the LC diet, the HC diet reduced the protein expression of Nrf2, and the protein expression of Nrf2 was increased by adding disodium fumarate in the HC diet (Figure 7H). Compared with the LC diet, the HC diet increased the fluorescence intensity of ROS in the mammary gland tissue of Hu sheep. However, the supplementation of disodium fumarate in the HC diet reversed the occurrence of this phenomenon (Figure 7I). These results indicate that the HC diet induced oxidative stress by inhibiting the Nrf2 signaling pathway in the mammary gland tissue of Hu sheep, and this phenomenon could be reversed by adding disodium fumarate in the HC diet.

## 4. Discussion

In order to rapidly improve the growth performance and lactation performance of animals, ruminants were fed excessive amounts of high concentrate. The high concentrates can be over-fermented in the rumen and result in SARA, which can damage the normal ruminal physiological metabolism of ruminants and induce a series of nutritional metabolic diseases, causing huge economic losses to the ruminant breeding industry [28,29,30]. Now, people have begun to pay attention to the prevention and control of ruminant disease, but due to the continuous strengthening of intensive feeding and excessive pursuit of profits, the incidence of SARA is still increasing year by year. Research investigations have shown that the incidence of SARA in cows in early and middle lactation ranges from 19% to 26% in dairy farms [28]. Therefore, the prevention and control of SARA to protect mammary gland health are of great significance to promote the healthy, efficiency, and sustainable development of the dairy industry.

Feeding cattle with highly soluble carbohydrates diets for a long time can cause rumen digestive dysfunction in ruminants. When the balance of microbial flora in the rumen is broken, which causes a large amount of volatile fatty acids to accumulate in the rumen, the pH value of the rumen will be reduced, and it is easy to instigate SARA. The rumen fluid pH value reflects the rumen fermentation status and can be used to determine whether SARA occurs in ruminants [1,2]. Our previous studies have also shown that a high-concentrate diet can induce subacute ruminal acidosis in goats [31]. In this experiment, the pH values of rumen fluid in the high-concentrate group were all lower than 5.6 after feeding, and the rumen pH was lower than 5.6 for more than 3 h a day. In addition, the HC diet elevated the concentration of LPS in rumen fluid and blood, which indicates that SARA was successfully induced in the Hu sheep. However, compared with the HC diet, the supplementation of disodium fumarate in the HC diet significantly increased the rumen pH. The rumen pH of the AHC diet was all above 5.6, which was close to that of the LC diet.

We found that, compared with LC diet, the HC diet tended to reduce the DMI of Hu sheep. However, the DMI of the HC group was not significantly different from that of the LC group. This was consistent with previous studies [32,33]. In this study, the decrease in rumen pH in the high-concentrate group may be due to the high concentration of soluble carbohydrates in the high-concentrate diet, which are rapidly fermented in the rumen, and thus large amounts of lactic acid and other organic acids are rapidly produced in the rumen. When excess acid is not absorbed or neutralized by the body in a timely manner, rumen pH is reduced, resulting in subacute rumen acidosis. However, disodium fumarate can promote rumen fermentation and increase the proliferation of cellulose decomposing bacteria and cellulose digestion so as to improve rumen pH levels under high-concentrate conditions [24,25,34]. This is consistent with the research of Mao et al. [26].

When the body is stimulated by external stimuli, a large number of unfolded proteins will be accumulated in the endoplasmic reticulum (ER), which will induce the dysfunction of the ER and the imbalance of the homeostasis of the cellular environment, thus inducing ER stress. ER stress is associated with the Ca^2+^ balance disorder [35]. When ER stress is prolonged, Ca^2+^ is released from the ER into the cytoplasm. IP3R is an important Ca^2+^ release channel on the surface of ER. It forms Ca^2+^ channel proteins with GRP75 and VDAC1 to mediate Ca^2+^ release from the endoplasmic reticulum and transport to mitochondria [36]. The mitochondrial calcium unidirectional transporter (MCU) plays an important role in the transfer of Ca^2+^ from the endoplasmic reticulum to the mitochondria. It can help Ca^2+^ to cross the inner mitochondrial membrane into the mitochondrial stroma. Studies have shown that the activation of the IP3R channel leads to a reduction in Ca^2+^ in the ER and an increase in Ca^2+^ in the mitochondria, thus leading to the apoptosis of hippocampal neurons [37]. Rapizzi et al. found that the overexpression of VDAC1 significantly increased the mitochondrial Ca^2+^ concentration in HeLa cells and skeletal muscle cells, while the knockout of VDAC1 significantly decreased the mitochondrial Ca^2+^ concentration [38]. Studies have shown that Ca^2+^ enters the mitochondrial matrix through MCU to control the accumulation of Ca^2+^ in the mitochondria and prevent the occurrence of mitochondrial Ca^2+^ overload [39,40]. Impaired MCU function can lead to mitochondrial Ca^2+^ overload, as well as the opening of the mitochondrial permeability conversion hole, triggering the release of cytochrome C, and ultimately leading to cell apoptosis [41]. Therefore, the maintenance of normal mitochondrial function depends on the appropriate concentration of calcium ions in mitochondria [42]. Our results indicate that HC diets induced ER stress in the mammary gland tissues of Hu sheep by up-regulating the mRNA and protein expression of ER stress-related proteins. In addition, the protein expression of IP3R, GRP75, VDAC1, and MCU could be regulated by feeding a HC diet. The results of immunofluorescence further show that the high-concentrate diet not only increased the fluorescence intensity of IP3R, GRP75, and VDAC1, but also promoted the interaction between IP3R and VDAC1 and between GRP75 and VDAC1. These results indicate that the high-concentrate diet could induce ER stress and activate the IP3R/GRP75/VDAC1-MCU signaling pathway in the mammary gland tissues of Hu sheep. Mitochondrial calcium overload is not conducive to the maintenance of fission fusion balance in mitochondria and can destroy the structure and function of mitochondria [43], which will further disrupt mitochondrial Ca^2+^ homeostasis and lead to mitochondrial damage. Studies demonstrated that the overexpression of OPA1 enhanced the process of mitochondrial division [44]. MFN2 knockdown inhibited the mitochondrial fusion process and significantly reduced the levels of MMP and ATP levels, which affected mitochondrial morphology and function [45,46,47]. Mollica et al. showed that obesity reduced the mRNA expressions of MFN1 and OPA1 and increased the mRNA expressions of Drp1 and Fis1 in the liver of mice, simultaneously causing a decrease in mitochondrial respiratory function [48]. We then examined the related indexes of mitochondrial genesis and mitochondrial dynamics. In this experiment, compared with the LC diet, the HC diet significantly increased the mRNA and protein expressions of Drp1, MFF, and Fis1 and decreased the mRNA and protein expressions of MFN2, MFN1, and OPA1 in the mammary gland tissues of Hu sheep. In addition, the high-concentrate diet also inhibited the protein expression of mitochondrial biogenesis-related proteins SIRT1, PGC-1α, and TFAM in the mammary gland tissues of Hu sheep. This suggests that the high-concentrate diet caused mitochondrial damage by inhibiting mitochondrial biogenesis and fusion and by promoting mitochondrial division. This phenomenon could be reversed when disodium fumarate was added into the high-concentrate diet.

Mitochondria represent an important organelle of energy metabolism in the body, as well as the main production sites of ROS [49]. Under normal conditions, the production and clearance of mitochondrial ROS maintain a dynamic balance [50]. When the mitochondrial structure and functional homeostasis are destroyed, the ROS level of the body far exceeds the antioxidant capacity of the cell. The excessive ROS and the damage of the antioxidant system will lead to an imbalance in the oxidation–antioxidant system and cause oxidative stress [51,52]. When animals are in SARA, it will lead to free radical metabolism disorder and the excessive production of ROS. The activity changes in SOD and the content changes in GSH and MDA in serum can reflect the antioxidant capacity of the body. MDA is the end product of lipid peroxides, which can cause cell damage. SOD and GSH can eliminate free radicals and inhibit the formation of MDA to maintain the oxidation balance in the body. Our previous study found that feeding high-concentrate diets significantly decreased the activity of T-AOC and GSH-Px in the serum of dairy cows, while the level of MDA tended to increase [33]. In this experiment, compared with the LC diet, the HC diet significantly decreased the activity of SOD and the content of GSH in the blood of Hu sheep and increased the content of MDA. However, adding disodium fumarate could increase the content of SOD and GSH in the blood of Hu sheep. This indicates that adding disodium fumarate to a high-concentrate diet improved the antioxidant defense mechanism system of the body and has a certain easing effect on the oxidative stress induced by the high-concentrate diet. In addition, compared with the HC diet, the AHC diet also increased the activity of SOD in the mammary gland tissue of Hu sheep, indicating that adding disodium fumarate to the high-concentrate diet improved the antioxidant function of the mammary gland tissue of Hu sheep and reduced the damage of free radicals to cells. The AHC diet reduced the content of MDA in mammary gland tissue, suggesting that the addition of disodium fumarate in the high-concentrate diet reduced the lipid peroxidation in mammary gland tissue, which might be due to the enhanced free-radical-scavenging ability of SOD and reduced the damage of free radicals to cell membranes. In addition, the addition of disodium fumarate in the high-concentrate diet significantly increased the content of GSH in mammary gland tissue, suggesting that the addition of disodium fumarate improved the redox state of mammary gland tissue and increased the antioxidant capacity of mammary gland tissue. Nrf2 plays an important role in the protective mechanism of oxidative stress. Yu et al. found that daidzein led to LPS-induced ROS reductions and increased the SOD activity in primary mouse hepatocytes through the up-regulation of the Nrf2 expression [53]. Therefore, we examined the gene and protein levels of Nrf2. Our results show that the mRNA and protein expression trends of Nrf2 in the LC, HC, and AHC diets were consistent in the mammary tissue of Hu sheep. Compared with the LC diet, the expression of Nrf2 in the HC diet was decreased, indicating that the HC diet caused oxidative damage to the mammary gland tissue of Hu sheep. However, after adding disodium fumarate into the high-concentrate diet, the expression of antioxidant-related genes was up-regulated by activating the Nrf2 signaling pathway. Thus, the antioxidant properties of mammary gland tissue in the SARA state were improved, and the mammary gland tissue of Hu sheep was protected from the attack of the high-concentrate diet, which may be one of the reasons why the addition of disodium fumarate can improve the mammary gland injury induced by the high-concentrate diet.

## 5. Conclusions

High concentrate diet induced SARA in Hu sheep by reducing rumen pH and increasing the concentration of LPS in blood and rumen fluid. In addition, the high-concentrate diet induced ER stress and mitochondrial damage by increasing the Ca^2+^ concentration in mammary gland tissue. The HC diet also induced oxidative stress in the mammary gland tissue of Hu sheep by inhibiting Nrf2. However, the supplementation of disodium fumarate at a daily dose of 10 g/sheep enhanced rumen bufferation by maintaining ruminal pH above 6 and reduced LPS concentration in ruminal fluid and blood. Thus, it alleviated the endoplasmic reticulum stress, oxidative stress, and mitochondrial damage of mammary gland tissue induced by the high-concentrate diet in Hu sheep.

## Figures and Tables

**Figure 1 antioxidants-12-00223-f001:**
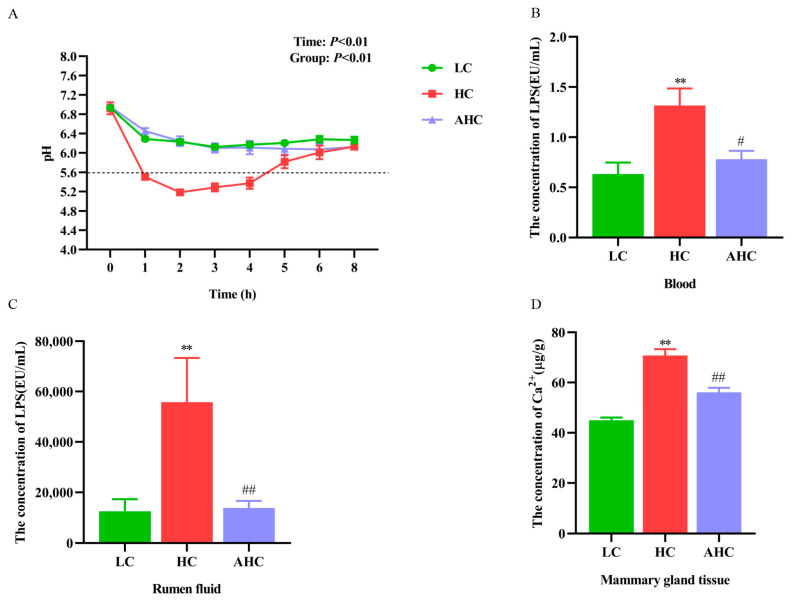
The effects of high-concentrate diet supplemented with disodium fumarate on rumen pH, the LPS content of blood, and rumen fluid, as well as Ca^2+^ content in mammary gland tissue. Rumen pH in the three diets (**A**); the content of LPS in blood (**B**) and rumen fluid (**C**); the content of Ca^2+^ in the mammary gland tissue of Hu sheep (**D**). The results are presented as the means ± SEM. ** *p* < 0.01 represents significant differences compared with the LC diet. ^#^
*p* < 0.05, ^##^
*p* < 0.01, represent significant differences compared with the HC diet.

**Figure 2 antioxidants-12-00223-f002:**
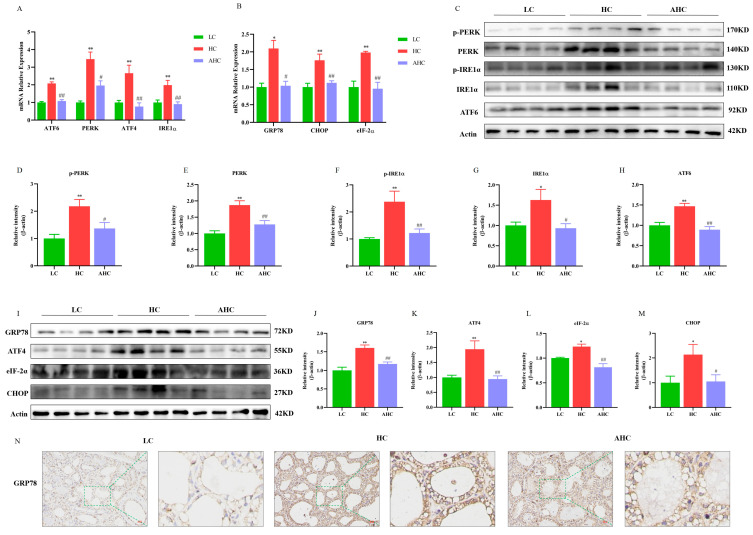
The effects of the high-concentrate diet supplemented with disodium fumarate on endoplasmic reticulum stress in the mammary gland tissue of Hu sheep. Relative mRNA expression of endoplasmic reticulum stress-related genes in the mammary gland tissue of Hu sheep (**A**,**B**); representative bands of endoplasmic reticulum stress-related proteins in the mammary tissue of Hu sheep (**C**,**I**). Relative protein abundances of PERK, p-PERK, IRE1α, p-IRE1α and ATF6 (**D**–**H**); relative protein abundances of GRP78, ATF4, eIF-2α, and CHOP (**J**–**M**); immunohistochemical results of GRP78 in mammary gland tissue of Hu sheep. The bar = 100 μm (**N**). Data are presented as the relative protein abundances relative to actin (means ± SEM), * *p* < 0.05, ** *p* < 0.01 represent significant differences compared with the LC diet. ^#^
*p* < 0.05, ^##^
*p* < 0.01 represent significant differences compared with the HC diet; immunohistochemical results of GRP78 in the mammary gland tissue of Hu sheep (200× magnification). The red bar = 100 μm (**N**) The green line represents zooming in on the selected area of the image.

**Figure 3 antioxidants-12-00223-f003:**
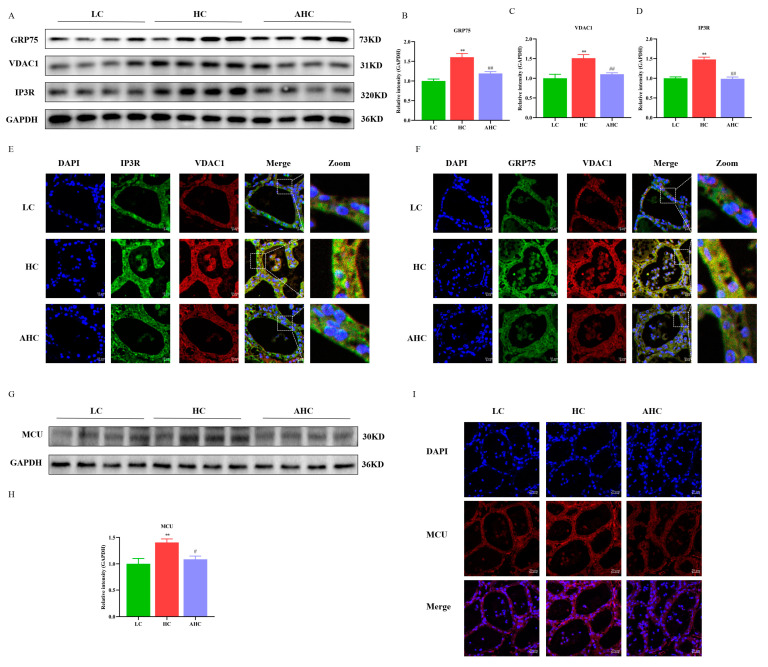
The effects of the high-concentrate diet supplemented with disodium fumarate on the IP3R-VDAC1-MCU of the mammary glands of Hu sheep. The protein expression of IP3R, GRP75, and VDAC1 in mammary gland tissue of Hu sheep (**A**–**D**); immunofluorescence results of IP3R, GRP75, and VDAC1 in the mammary tissue of Hu sheep. The bar = 10 μm (**E**,**F**). The white line represents zooming in on the selected area of the image; the protein expression of MCU in the mammary gland tissue of Hu sheep (**G**,**H**); immunofluorescence results of MCU in the mammary gland tissue of Hu sheep. The bar = 20 μm (**I**). Data are presented as the relative protein abundances relative to GAPDH (means ± SEM), ** *p* < 0.01 represents significant differences compared with the LC diet. ^#^
*p* < 0.05, ^##^
*p* < 0.01 represent significant differences compared with the HC diet.

**Figure 4 antioxidants-12-00223-f004:**
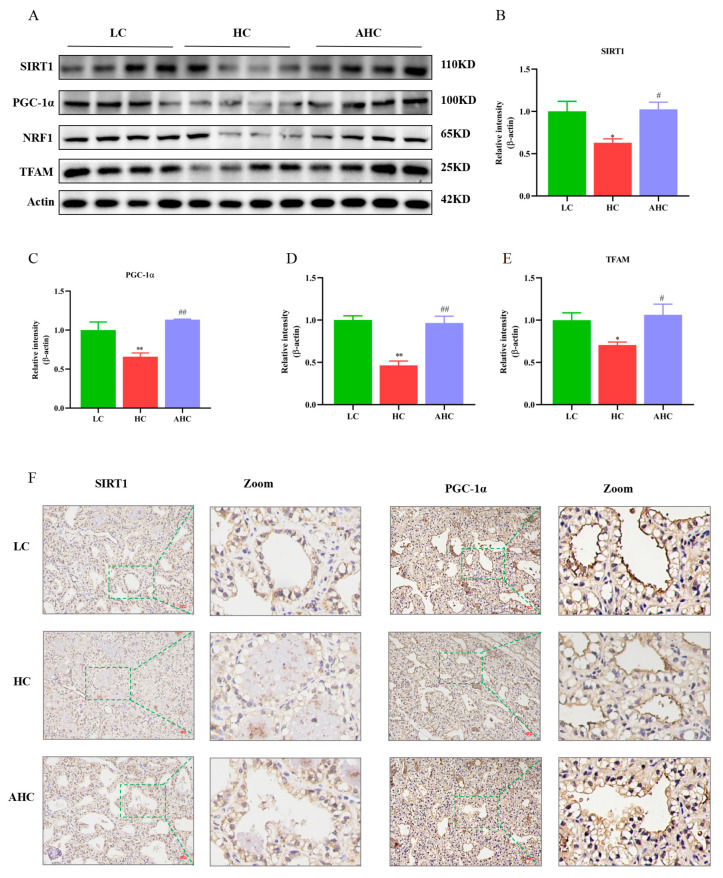
The effects of the high-concentrate diet supplemented with disodium fumarate on mitochondrial biogenesis in the mammary gland tissue of Hu sheep. Mitochondrial biogenesis-related protein expression (**A**–**E**). Data are presented as the relative protein abundances relative to actin (means ± SEM). * *p* < 0.05, ** *p* < 0.01 represent significant differences compared with the LC diet. ^#^
*p* < 0.05, ^##^
*p* < 0.01 represent significant differences compared with the HC diet. Immunohistochemical results of SIRT1 and PGC-1α in the mammary gland tissue of Hu sheep. The bar = 100 μm (**F**). The green line represents zooming in on the selected area of the image.

**Figure 5 antioxidants-12-00223-f005:**
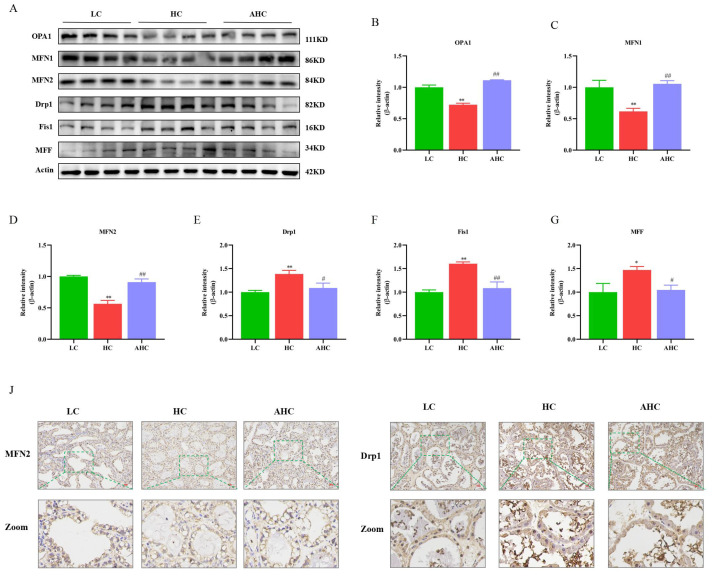
The effects of disodium fumarate on mitochondrial dynamics in the mammary glands of Hu sheep induced by the high-concentrate diet. Representative bands of the proteins associated with mitochondrial fusion and division (**A**); quantified volumes of specific bands (**B**–**G**); data are presented as the relative protein abundances relative to actin (means ± SEM), * *p* < 0.05, ***p* < 0.01 represent significant differences compared with the LC diet. ^#^
*p* < 0.05, ^##^
*p* < 0.01 represent significant differences compared with the HC diet. Immunohistochemical results of MFN2 (**H**) and Drp1 (**I**) in the mammary gland tissue of Hu sheep. The bar = 100 μm. The green line represents zooming in on the selected area of the image.

**Figure 6 antioxidants-12-00223-f006:**
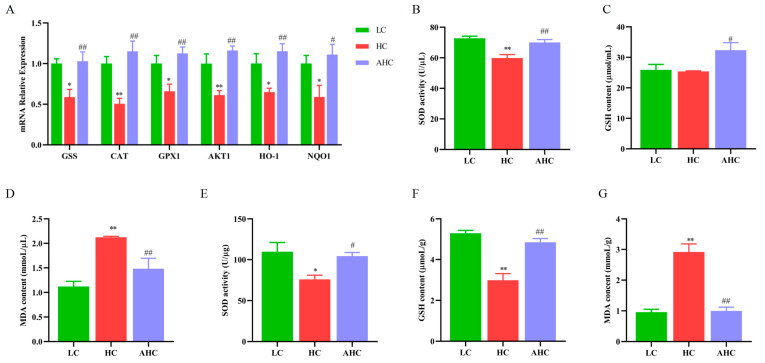
The effects of disodium fumarate supplementation in the high-concentrate diet on oxidative stress-related genes in mammary gland tissue and oxidative stress-related indicators in plasmid and mammary gland tissue. The relative mRNA abundance of the genes related to oxidative stress in the mammary gland tissue of Hu sheep (**A**); the activity of SOD (**B**), the concentration of MDA and GSH in blood (**C**,**D**) and mammary gland tissue (**E**–**G**) in Hu sheep. Data are presented as the relative protein abundance relative to actin (means ± SEM), * *p* < 0.05, ** *p* < 0.01 represent significant differences compared with the LC diet. ^#^ *p* < 0.05, ^##^ *p* < 0.01 represent significant differences compared with the HC diet.

**Figure 7 antioxidants-12-00223-f007:**
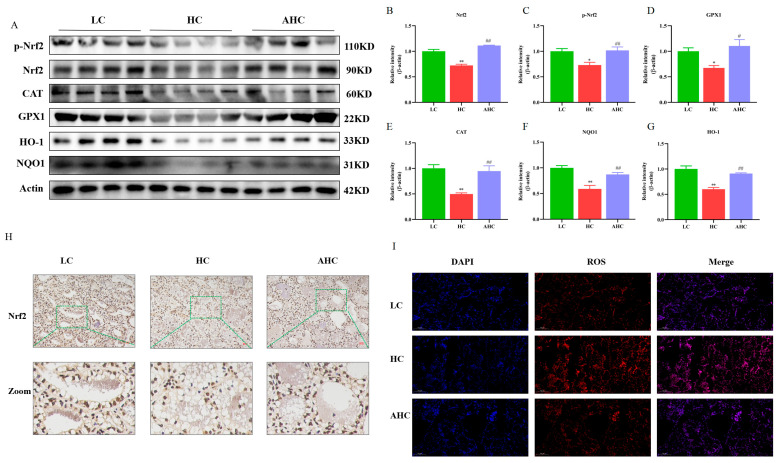
Disodium fumarate relieved the oxidative stress of mammary gland tissue induced by the the high-concentrate diet. The protein expression of oxidative stress-related proteins (**A**–**G**). Data are presented as the relative protein abundances relative to actin (means ± SEM), * *p* < 0.05, ** *p* < 0.01 represent significant differences compared with LC diet. ^#^
*p* < 0.05, ^##^
*p* < 0.01 represent significant differences compared with HC diet; immunofluorescence results of Nrf2, the red bar = 100 μm (**H**). The green line represents zooming in on the selected area of the image. The fluorescence of ROS in the mammary tissue of Hu sheep. The bar = 100 μm (**I**).

**Table 1 antioxidants-12-00223-t001:** Ingredients and nutrient contents of the experimental diets (DM basis,%).

Items	Diets (Content) ^3^	
	LC	HC
Maize	19.28	46.86
Soybean meal	4.20	9.80
Wheat bran	2.70	8.10
Rapeseed meal	1.20	2.80
Peanut vine	35.00	15.00
Cron stover silage	35.00	15.00
Limestone	0.42	0.82
NaCl	0.50	0.50
Premix ^1^	0.50	0.50
CaHPO_4_	1.20	0.62
Total	100.00	100.00
Nutrient levels ^2^		
CP	9.75	13.53
Ca	0.60	0.60
P	0.40	0.40
NDF	40.66	24.41

Note: ^1^ The premix provided the following per kg of diet: vitamin A, 4000 IU, vitamin D3, 400 IU, vitamin E, 20,000 IU, FeSO_4_, 69.06 mg, CuSO_4_, 17.6 mg, K_2_SO_4_, 31.70 mg, ZnSO_4_, 57.14 mg, MnSO_4_, 44.03 mg, CoCl_2_, 0.25 mg, Na_2_SeO_3_, 8.95 mg, NaHCO_3_, 3740.91 mg. ^2^ Nutrient levels were estimated values. ^3^ LC: a low concentrate diet, concentrate: forage = 30:70; HC: a high concentrate diet, concentrate: forage = 70:30; AHC: a high concentrate with disodium fumarate diet (each sheep was given an additional 10 g disodium fumarate/day).

**Table 2 antioxidants-12-00223-t002:** Supplemental disodium fumarate effects in dry matter intake of Hu sheep.

Item	Diets			*p* Value
	LC	HC	AHC	
DMI (kg/d)	1.53 ± 0.11	1.39 ± 0.17	1.49 ± 0.15	*p* > 0.05

## Data Availability

All data are comprised within this manuscript.

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
