# Peer review of "Disodium Fumarate Alleviates Endoplasmic Reticulum Stress, Mitochondrial Damage, and Oxidative Stress Induced by the High-Concentrate Diet in the Mammary Gland Tissue of Hu Sheep"

_antioxidants, 2023, doi:10.3390/antiox12020223_

Round 1

Reviewer 1 Report

Comments and Suggestion for authors :

This manuscript describes that fumarate supplementation alleviates endoplasmic reticulum and oxidative stress during SARA in Hu sheep fed a high-concentration diet. A couple of comments are given below to further strengthen the quality of the manuscript.

Materials and Methods

Line 119: How is fumarate added to the feed (mix or top dressing)? Also, what is the basis for adding 10g of fumarate? In this manuscript, the amount of feed provided was 110% of the previous day's intake. Is there a correlation between the amount of feed intake and the amount of fumarate provided? Please explain.

Line 124: Is it AHC or CHC? Please check the treatment name throughout the manuscript.

Line 126: Are the experimental diets TMR? If so, it should described as TMR on line 126 or supplementary Table 1.

Line 143: Indicate 0h in the sampling period.

Line 144: pH mete -> pH meter. Please carefully check the spellings throughout the manuscript.

Results

When the pH of rumen fluid ranges from 5.6 to 5.8, it leads to a drop in acetate-to-propionate ratio, fiber digestibility, and DMI. Have the authors checked the DMI? Please include in the results and discussion part.

Fig.1A : Even though you fed different diets for 8 weeks, the pH appears to be similar at 0h. I assume that since the sheeps were fed different diets, the pH should vary from each animal. Please explain your result.

Conclusion

Line 520: inhibitede -> inhibited

Figures

Figures should be in high resolution and the font size is too small to read.

Check figure label format for consistency. (Lines 252, 277, 303, 321, 343, 377)

Author Response

Response to Reviewer 1 Comments

Dear Reviewer:

We are very grateful to Reviewers for reviewing the paper so carefully. First of all, we are deeply sorry for the lack of detailed description and inappropriate expression in the article. In the future, we will check and check carefully to avoid the occurrence of low-level errors.

We have carefully considered the suggestion of Reviewers, and tried our level best to address all the points raised by worthy reviewers to make the manuscript more convincing. The detailed response against each point asked by reviewers is highlighted (high light) in the main manuscript file and appended below:

This manuscript describes that fumarate supplementation alleviates endoplasmic reticulum and oxidative stress during SARA in Hu sheep fed a high-concentration diet. A couple of comments are given below to further strengthen the quality of the manuscript.

Materials and Methods

Point 1: Line 119: How is fumarate added to the feed (mix or top dressing)? Also, what is the basis for adding 10g of fumarate? In this manuscript, the amount of feed provided was 110% of the previous day's intake. Is there a correlation between the amount of feed intake and the amount of fumarate provided? Please explain.

Response 1: Dear reviewer, thank you for your suggestion. Because the amount of disodium fumarate was too small for each sheep, it was necessary to ensure that each sheep in AHC group received enough disodium fumarate (10g) per day. Therefore, we mixed disodium fumarate with a certain amount of concentrate (to ensure that the Hu sheep could completely consume disodium fumarate) and fed it to the experimental sheep.

About the basis for adding 10g of fumarate: Before the experiment, we reviewed the relevant literature. Yu et al. showed that Supplementing the diet with Fumarate addition (6 g/head per day) had significant effects on rumen microbial fermentation by decreasing ammonia and branched-chain VFA, and by increasing acetate and propionate, and NDF digestion [1]. Disodium fumarate supplementation at a level of 20 g/d increases the ruminal fluid pH from 6.74 to 6.94, and alters microbial populations in Hu sheep fed a high-forage diet, indicating the positive effects of disodium fumarate on rumen fermentation [2]. Study has demonstrated the supplementation of 8 mM disodium fumarate decreases the acetate: propionate ratio from 1.74 to 1.40, and increases the final pH from 5.99 to 6.07 in the ruminal contents of Jersey steer [3]. Therefore, based on the above studies, we selected a middle value (10 g/d) of disodium fumulate to investigate whether adding 10g/d of disodium fumulate to high-concentrate diet could alleviate endoplasmic reticulum stress, mitochondrial damage and oxidative stress induced by high-concentrate diet in mammary gland tissue of Hu sheep.

We provided 110% of the previous day's feed to ensure that each animal was able to meet ad libitum feeding. Avoid the phenomenon of not eating enough. Because we mixed disodium fumarate with a small amount of concentrate, and then fed it to the experimental sheep in advance. Therefore, animal feed intake did not affect the intake of disodium fumarate.

References:

[1] Yu CW, Chen YS, Cheng YH, Cheng YS, Yang CM, Chang CT. Effects of fumarate on ruminal ammonia accumulation and fiber digestion in vitro and nutrient utilization in dairy does. J Dairy Sci. 2010 Feb;93(2):701-10. doi: 10.3168/jds.2009-2494.

[2] Zhou YW, McSweeney CS, Wang JK, Liu JX. Effects of disodium fumarate on ruminal fermentation and microbial communities in sheep fed on high-forage diets. Animal. 2012 May;6(5):815-23. doi: 10.1017/S1751731111002102.

[3] Callaway TR, Martin SA. Effects of organic acid and monensin treatment on in vitro mixed ruminal microorganism fermentation of cracked corn. J Anim Sci. 1996 Aug;74(8):1982-9. doi: 10.2527/1996.7481982x.

Point 2: Line 124: Is it AHC or CHC? Please check the treatment name throughout the manuscript.

Response 2: We appreciate it very much for this good suggestion. We are very sorry for our incorrect writing and it is rectified in the article. The fumarate addition diet is AHC diet.

Point 3: Line 126: Are the experimental diets TMR? If so, it should described as TMR on line 126 or supplementary Table 1.

Response 3: The diets in the experiment were not TMR. The diets in the experiment ware processed by ourselves according to the diet table (Table1).

Point 4: Line 143: Indicate 0h in the sampling period.

Response 4: We appreciate it very much for this good suggestion, and we have done it according to your ideas

Point 5: Line 144: pH mete -> pH meter. Please carefully check the spellings throughout the manuscript.

Response 5: We appreciate it very much for this good suggestion. We are very sorry for our incorrect writing. The incorrect spelling has been corrected in the article.

Results

Point 6: When the pH of rumen fluid ranges from 5.6 to 5.8, it leads to a drop in acetate-to-propionate ratio, fiber digestibility, and DMI. Have the authors checked the DMI? Please include in the results and discussion part.

Response 6: We recorded the feed intake of the experimental sheep daily. The results of the study showed that compared with the low concentrate diet, the high concentrate diet tended to reduce the feed intake of Hu sheep. However, there was no significant difference in DMI among the three groups. This was consistent with previous studies [1,2]. Their results showed that compared with the low concentrate diet, the high concentrate diet did not affected the DMI of dairy cows. We have added data on feed intake to the results of the article, and it is discussed accordingly in the discussion.

References:

[1] Li L, Cao Y, Xie Z, Zhang Y. A High-Concentrate Diet Induced Milk Fat Decline via Glucagon-Mediated Activation of AMP-Activated Protein Kinase in Dairy Cows. Sci Rep. 2017. 13;7:44217. http://doi: 10.1038/srep44217.

[2] Ma N, Abaker JA, Wei G, Chen H, Shen X, Chang G. A high-concentrate diet induces an inflammatory response and oxidative stress and depresses milk fat synthesis in the mammary gland of dairy cows. J Dairy Sci. 2022. 105(6):5493-5505. http://doi: 10.3168/jds.2021-21066.

Point 7: Fig.1A: Even though you fed different diets for 8 weeks, the pH appears to be similar at 0h. I assume that since the sheeps were fed different diets, the pH should vary from each animal. Please explain your result.

Response 7: High concentrate diets with high carbohydrate content such as starch and soluble sugars can lead to accelerated rumen fermentation rate. When volatile fatty acids exceed the buffering capacity and absorption capacity of the rumen, and the rumen cannot effectively neutralize these organic acids in time, the pH value of the rumen will decrease. When the ruminal pH is below 5.6 for more than 3 h, which leads to subacute ruminal acidosis. Our experiment was induced subacute ruminal acidosis in Hu sheep. When Hu sheep are fed a high-concentrate diet, it will be rapidly degraded by rumen microorganisms, resulting in a large amount of volatile fatty acids, which can lead to a rapid decline in rumen pH for a period of time. When the easily fermentable carbohydrates in the diet are catabolized, saliva enters the rumen at the time of rumination, neutralizing organic acids and increasing rumen pH. The pH in the rumen will slowly rise and return to normal values. The rumen pH was in a relatively stable state. That is, after feeding high concentrate diet, the trend of ruminal pH of Hu sheep was first decreased, and then gradually increased with the extension of time. After feeding at 16:00 on the previous day, the feed was digested and metabolized overnight in the rumen of the animals. By the time we measured the rumen pH at 9:00 on the next day, the feed had been almost completely digested and metabolized by the rumen. So ruminal pH appears to be similar at 0 h in the Hu sheep of the three diets. This is similar to previous studies [1,2].

You are right that the difference between the individual sheep is inevitable. But even if there are differences between individuals, their rumen pH is still within a certain range, so there is little difference in rumen pH between groups.

References:

[1] Dong H, Wang S, Jia Y, Ni Y, Zhang Y, Zhuang S, Shen X, Zhao R. Long-term effects of subacute ruminal acidosis (SARA) on milk quality and hepatic gene expression in lactating goats fed a high-concentrate diet. PLoS One. 2013. 23;8(12):e82850. doi: 10.1371/journal.pone.0082850.

[2] Mu YY, Qi WP, Zhang T, Zhang JY, Mao SY. Gene function adjustment for carbohydrate metabolism and enrichment of rumen microbiota with antibiotic resistance genes during subacute rumen acidosis induced by a high-grain diet in lactating dairy cows. J Dairy Sci. 2021. 104(2):2087-2105. doi: 10.3168/jds.2020-19118.

Conclusion

Point 8: Line 520: inhibitede -> inhibited

Response 8: We appreciate it very much for this good suggestion. We are very sorry for our incorrect writing. The incorrect spelling has been corrected in the article.

Figures

Point 9: Figures should be in high resolution and the font size is too small to read.

Check figure label format for consistency. (Lines 252, 277, 303, 321, 343, 377)

Response 9: We appreciate it very much for this good suggestion. We made corresponding adjustments to the pictures that were not clear in the article. The figure label format has been modified according to your requirements

Reviewer 2 Report

Specific comments.

Introduction is too long, with irrelevant parts and many problems in English language.

L40-42: rephrase needed is not comprehensive

L48-54: not comprehensive

L74-78: not comprehensive, it is suggested to be re-written

L76-78: this claim is not justified as a conclusion or not supported by a relative literature reference. Moreover, it claims the opposite in the next sentence (L79-80), where authors aim to prove it in the present work

L88-89:rephrase needed is not comprehensive

L100-111: this part is supposed to present the aims of the study, the research hypothesis and the action to prove it. It is not clear. I cannot see what actually is conclusion? Results?

Study design concerning animals is not appropriate and it is not explained or presented.

It is clear when experiment begins and when finishes

L120: mid lactation sheep in 15 days post partum? How long is duration of lactation in Hu sheep 30 days? The whole feeding trial is supposed to last 11 weeks, if I understood well from the rest information

L124: The fumarate addition diet is referred elsewhere as AHC. This CHC abb. here is confusing.

Even in the supplemental table is not clear what are the animals fed. Percentage of ingredients in the diet are available, but nowhere the exact amount of food is presented. The study design is supposed to feed 3 different dies in order to induce SARA and I could find what exactly these animals were fed! How did the authors harvested all the claimed results from SARA, when we do not know how the animals were fed. It is serious omission. Also monensin is a routine inclusion in sheep feeding in china? It would claim bad antibiotic practices to feed it in healthy sheep ration.

Additionally, how authors decided to feed 10 gr of disodium fumarate/ day? There is no relative literature on this. Did they make previous trials? It is at least amazing for me that they found the exact amount (and not 5 or 25grams for example) as it can be concluded by the absolutely positive results presented in the study with this dose.

It would also be necessary to mention that all animals were equally allocated in 3 groups, reader cannot guess so.

L130: 2 times water

L131: authors claim to give 110% of previous intake feed, but not even here provide the amount of feed, how much these animals per group were fed??

It is very important but also not mentioned nowhere: where these lactating animals (midlactation as mentioned above) actually milking? Authors claim that the feeding experiment lasted 3+8=11 weeks. Were these animals milking? If not it should be clearly mentioned and provide details about drying process. If yes a lot of information is needed. Authors organized a study to see the effect of SARA (without providing exact feeding) on mammary gland health and status and do not tell us whether animals where milking! How can milking, mastitis and udder health cannot be affected by other parameters, but only by feeding? Authors claim Ca concentrations difference and don’t say anything about milk production! I think this is a serious fault.

I would also expect to have some clinical signs like inappetence in the group of SARA induction, since 8 weeks of High concentrate feeding is due time for this. In general it seems that authors made a lot of laboratory work after slaughtering the animal, but all the 11 weeks feeding/animal part is like a ghost.

L139: other samples like? Udder parenchyma samples?

L143: Rumen fistula! Authors should provide more details about this. Did they fistulated all 18 sheep? How? When? Fistulation is a surgical procedure that needs to be completed much earlier than the experimental period. So authors make rumen fistulation in 18 sheep for just 3 days of sampling? (this can be assumed from the not clear description-3 last days of the last week of experiment). It poses a serious question on animal welfare this procedure.

L143: earlier (L131) is mentioned that animal were fed twice daily (8-16:00), so which feeding is supposed to be here? It needs clarification

L145: last day remains not clear. It is the last day of the 8 weeks trial or the next? Again experimental scheme is not clear

L145-146: it needs clarification. I do not think that authors actually collected blood in an empty stomach!

L242: AHC abbreviation is not mentioned again in next, earlier was CHC, it is confusing

L387-388: it is not comprehensive

L406-407: it is not comprehensive

L501-502: it is not comprehensive

Author Response

Response to Reviewer 2 Comments

Dear Reviewer:

Thank you for your valuable comments about our article. Please file authors' response to your comments in the attached pdf 

Reviewer 3 Report

The purpose of this study was to investigate the effects of disodium fumarate included as a feed additive in high-concentrate diets as a preventive of the presentation of low rumen pH (SARA) measuring it effectivity on endoplasmic reticulum stress, mitochondrial damage and oxidative stress in mammary gland tissue of Hu Sheep. Thus, the subject falls within the general scope of the journal. This manuscript reports on a topic pertinent to ruminant nutrition. The manuscript is well written and organized. The data presented are good and sufficient to provide very good quality information to contrast the hypothesis raised. Material and methods are exhaustively described and statistical methods used are adequate. Results are well developed and discussions are sustained adequately. It is an excellent paper that contributes to the knowledge of the topic that it deals with. However, a few flaws must be corrected before I considered to be published: 

L16: Specify the dose of fumarate used.

L29: Conclusion must be rewrite. Redirect your conclusion supporting the hypothesis! (Hint): Supplementation of disodium fumarate at daily dose of 10 g/sheep enhanced rumen bufferation maintaining ruminal pH above 6. and reduced LPS concentration in ruminal fluid and blood This reaction avoided the negative effect observed to non-supplemented sheep that were fed with a high concentrate diet which consisted in..

L48: Please, don’t confuse “rumen acidosis” with “subacute ruminal acidosis (SARA)”. According to your experimental design, high concentrate treatment favors SARA not ruminal acidosis. Please here and through the document define it as subacute ruminal acidosis.

L47: Please include that: SARA is characterized by daily episodes (> 2 h) of low ruminal pH in the range of 5.2 and 6 and it’s difficult to diagnose because of the absence of overt clinical signs it's very difficult to detect in a short period of time (Krause and Otzel, 2006; Nagaraja and Lechtenberg, 2007; Li et al., 2013)

L48-51: Please sustain this statement with at least one reference. For example, 20% of incidence of SARA in Dairy Herds have been determined by Kleen et al (2013)

It has been determined SARA incidences of 20% in dairy herds and 35% in feedlot cattle (Kleen et al. 2013; Attia, 2016)

L88: Include “feed” (i.e.) Therefore, fumaric acid is harmful to rumen fermentation of ruminants feed with high concentrate diets

L103: Hypothesis is not well posed. Please rewrite the hypothesis as: We hypothesized that supplementation (10 g/day) of disodium fumarate in Hu Sheep feeding with a high-concentrate diet prevents the drop in rumen pH to risk levels for SARA presentation, avoiding endoplasmic reticulum stress, oxidative stress and mitochondrial damage in mammary gland tissue.

L120: All sheep had a ruminal cannula? Please specify. Sheep were allocated in individual pens or in a group pen? Specify pen characteristics

L122: “Concentrate” is not equivalent to “high soluble carbohydrates”. The effective NDF is an important factor as well. Therefore, is essential to the readers that describe the concentrate and source of forage used in the paper (not in a supplementary file). Please rewrite this part (Hint) A concentrate (70% cracked corn, 17% DDGS, 10% soybean meal, 3% mineral supplement) and forage (rice straw) was combined to obtain a low-concentrate diet (30:70) and high concentrate diet (70:30). Treatments were as follows:  1) low concentrate (LC) diet, 2) high concentrate (HC) diet, and 3) high concentrate (AHC) supplemented with disodium fumarate at daily dose of 10 g sodium fumarate/sheep.

L124: How the 10 g of disodium fumarate was offered to sheep? Specify. CHC or AHC? Please, be congruent with the acronyms used to define the treatments

L129-131: Please, move this paragraph to the 2.2 Animals and diets

L128: Reorder the paragraphs as follow: 1) Ruminal fluid, blood collection and the LPS concentration, and finally slaughter.

L138: gland samples were collected from all sheep? Specify

L142: Rumen fluid of all sheep was sampled…

L144: Rumen pH was measured immediately after filtration? Specify

L146: Blood was collected from the jugular vein of all sheep

L236: Rewrite as: 3.1. Supplemental disodium fumarate effects on ruminal pH, ruminal fluid and blood LP concentration, and calcium content in mammary gland tissue.

Figures are to small! Please resized all figures on order to be more easy interpretation.

L258: Rewrite as: 3.2. Supplemental disodium fumarate effects on endoplasmic reticulum stress indicators in mammary gland tissue.

L287: Rewrite as: 3.3. Supplemental disodium fumarate effects on IP3R-VDAC1-MCU expression in mammary gland tissue.

L311: Rewrite as: 3.4. Supplemental disodium fumarate effects on mitochondrial biogenesis in mammary gland tissue.

L327: Rewrite as: 3.5. Supplemental disodium fumarate effects on mitochondrial dynamics in mammary gland

L351: Rewrite as: 3.6. Supplemental disodium fumarate effects on oxidative stress of mammary gland tissue.

L390: Please sustain this statement with at least one reference.

L394: Please sustain this statement with at least one reference.

L397: Feeding cattle with high-soluble carbohydrates diets

L398: Good and bad bacteria? Please use more proper terms. In the rumen environment, instead of “body”

Conclusion

As mentioned above (L29), conclusion must be rewrite. Redirect your conclusion supporting the hypothesis!

Author Response

Response to Reviewer 3 Comments

Dear Reviewer:

Thank you very much for your recognition of our work. Thank you for your valuable comments about our article, which makes our article more perfect. We have carefully considered the suggestion of Reviewers, and tried our level best to address all the points raised by worthy reviewers to make the manuscript more convincing. The detailed response against each point asked by reviewers is highlighted (high light) in the main manuscript file and appended below:

The purpose of this study was to investigate the effects of disodium fumarate included as a feed additive in high-concentrate diets as a preventive of the presentation of low rumen pH (SARA) measuring it effectivity on endoplasmic reticulum stress, mitochondrial damage and oxidative stress in mammary gland tissue of Hu Sheep. Thus, the subject falls within the general scope of the journal. This manuscript reports on a topic pertinent to ruminant nutrition. The manuscript is well written and organized. The data presented are good and sufficient to provide very good quality information to contrast the hypothesis raised. Material and methods are exhaustively described and statistical methods used are adequate. Results are well developed and discussions are sustained adequately. It is an excellent paper that contributes to the knowledge of the topic that it deals with. However, a few flaws must be corrected before I considered to be published:

Point 1: L16: Specify the dose of fumarate used.

Response 1: Dear reviewer, thank you for your suggestion. Each sheep was given an additional 10 g disodium fumarate/day. The article has been supplemented in accordance with your requirements. The specific contents were also added to the manuscript. Please see the revised manuscript.

Point 2: L29: Conclusion must be rewrite. Redirect your conclusion supporting the hypothesis! (Hint): Supplementation of disodium fumarate at daily dose of 10 g/sheep enhanced rumen bufferation maintaining ruminal pH above 6. and reduced LPS concentration in ruminal fluid and blood This reaction avoided the negative effect observed to non-supplemented sheep that were fed with a high concentrate diet which consisted in.

Response 2: Dear reviewer, thank you for your suggestion. We have modified it according to your suggestion. The specific contents were also added to the manuscript. Please see the revised manuscript.

Point 3: L48: Please, don’t confuse “rumen acidosis” with “subacute ruminal acidosis (SARA)”. According to your experimental design, high concentrate treatment favors SARA not ruminal acidosis. Please here and through the document define it as subacute ruminal acidosis.

Response 3: Dear reviewer, thank you for your suggestion, and I am very sorry for the wrong description. We have revised the article accordingly.

Point 4: L47: Please include that: SARA is characterized by daily episodes (> 2 h) of low ruminal pH in the range of 5.2 and 6 and it’s difficult to diagnose because of the absence of overt clinical signs it's very difficult to detect in a short period of time (Krause and Otzel, 2006; Nagaraja and Lechtenberg, 2007; Li et al., 2013)

Response 4: Dear reviewer, thank you for your suggestion. We have revised the article according to your request.

Point 5: L48-51: Please sustain this statement with at least one reference. For example, 20% of incidence of SARA in Dairy Herds have been determined by Kleen et al (2013)

It has been determined SARA incidences of 20% in dairy herds and 35% in feedlot cattle (Kleen et al. 2013; Attia, 2016)

Response 5: Dear reviewer, thank you for your suggestion. We have revised the article according to your request.

Point 6: L88: Include “feed” (i.e.) Therefore, fumaric acid is harmful to rumen fermentation of ruminants feed with high concentrate diets

Response 6: Dear reviewer, thank you for your suggestion. We have revised the article according to your request.

Point 7: L103: Hypothesis is not well posed. Please rewrite the hypothesis as: We hypothesized that supplementation (10 g/day) of disodium fumarate in Hu Sheep feeding with a high-concentrate diet prevents the drop in rumen pH to risk levels for SARA presentation, avoiding endoplasmic reticulum stress, oxidative stress and mitochondrial damage in mammary gland tissue.

Response 7: Dear reviewer, thank you for your suggestion. We have revised the article according to your request.

Point 8: L120: All sheep had a ruminal cannula? Please specify. Sheep were allocated in individual pens or in a group pen? Specify pen characteristics.

Response 8: Dear reviewer, thank you for your suggestion. We have revised the article according to your request. The sheep were kept under identical conditions and individually feed in an indoor pen (0.97 m × 2.82 m). 

Point 9: L122: “Concentrate” is not equivalent to “high soluble carbohydrates”. The effective NDF is an important factor as well. Therefore, is essential to the readers that describe the concentrate and source of forage used in the paper (not in a supplementary file). Please rewrite this part (Hint) A concentrate (70% cracked corn, 17% DDGS, 10% soybean meal, 3% mineral supplement) and forage (rice straw) was combined to obtain a low-concentrate diet (30:70) and high concentrate diet (70:30). Treatments were as follows:  1) low concentrate (LC) diet, 2) high concentrate (HC) diet, and 3) high concentrate (AHC) supplemented with disodium fumarate at daily dose of 10 g sodium fumarate/sheep.

Response 9: Dear reviewer, thank you for your suggestion. We transferred the dietary composition table from the supplementary table to the main text in the article. The diet composition has been supplemented as suggested by you.

Point 10: L124: How the 10 g of disodium fumarate was offered to sheep? Specify. CHC or AHC? Please, be congruent with the acronyms used to define the treatments.

Response 10: Dear reviewer, thank you for your suggestion. Because the amount of disodium fumarate was too small for each sheep, it was necessary to ensure that each sheep in AHC group received enough disodium fumarate (10 g) per day. Therefore, we mixed disodium fumarate with a certain amount of concentrate (to ensure that the Hu sheep could completely consume disodium fumarate) and fed it to the experimental sheep. We are very sorry for our incorrect writing and it is rectified in the article.

Point 11: L129-131: Please, move this paragraph to the 2.2 Animals and diets.

Response 11: Dear reviewer, thank you for your suggestion. We have revised the article according to your request.

Point 12: L128: Reorder the paragraphs as follow: 1) Ruminal fluid, blood collection and the LPS concentration, and finally slaughter.

Response 12: Dear reviewer, thank you for your suggestion. We have revised the article according to your request.

Point 13: L138: gland samples were collected from all sheep? Specify.

Response 13: Dear reviewer, thank you for your suggestion. We have revised the article according to your request.

Point 14: L142: Rumen fluid of all sheep was sampled…

Response 14: Dear reviewer, thank you for your suggestion. We have revised the article according to your request.

Point 15: L144: Rumen pH was measured immediately after filtration? Specify.

Response 15: Dear reviewer, thank you for your suggestion. We have revised the article according to your request.

Point 16: L146: Blood was collected from the jugular vein of all sheep.

Response 16: Dear reviewer, thank you for your suggestion. We have revised the article according to your request.

Point 17: L236: Rewrite as: 3.1. Supplemental disodium fumarate effects on ruminal pH, ruminal fluid and blood LP concentration, and calcium content in mammary gland tissue.

Figures are to small! Please resized all figures on order to be more easy interpretation.

Response 17: Dear reviewer, thank you for your suggestion. We have revised the article according to your request. We reformatted Figures 1 to make sure you see the numbers in the figure.

Point 18: L258: Rewrite as: 3.2. Supplemental disodium fumarate effects on endoplasmic reticulum stress indicators in mammary gland tissue.

Response 18: Dear reviewer, thank you for your suggestion. We have revised the article according to your request.

Point 19: L287: Rewrite as: 3.3. Supplemental disodium fumarate effects on IP3R-VDAC1-MCU expression in mammary gland tissue.

Response 19: Dear reviewer, thank you for your suggestion. We have revised the article according to your request.

Point 20: L311: Rewrite as: 3.4. Supplemental disodium fumarate effects on mitochondrial biogenesis in mammary gland tissue.

Response 20: Dear reviewer, thank you for your suggestion. We have revised the article according to your request.

Point 21: L327: Rewrite as: 3.5. Supplemental disodium fumarate effects on mitochondrial dynamics in mammary gland.

Response 21: Dear reviewer, thank you for your suggestion. We have revised the article according to your request.

Point 22: L351: Rewrite as: 3.6. Supplemental disodium fumarate effects on oxidative stress of mammary gland tissue.

Response 22: Dear reviewer, thank you for your suggestion. We have revised the article according to your request.

Point 23: L390: Please sustain this statement with at least one reference.

Response 23: Dear reviewer, thank you for your suggestion. The article has been supplemented in accordance with your requirements.

Point 24: L394: Please sustain this statement with at least one reference.

Response 24: Dear reviewer, thank you for your suggestion. The article has been supplemented in accordance with your requirements.

Point 25: L397: Feeding cattle with high-soluble carbohydrates diets.

Response 25: Dear reviewer, thank you for your suggestion. We have revised the article according to your request.

Point 26: L398: Good and bad bacteria? Please use more proper terms. In the rumen environment, instead of “body”.

Response 26: Dear reviewer, thank you for your suggestion. We have revised the article according to your request.

Conclusion

Point 27: As mentioned above (L29), conclusion must be rewrite. Redirect your conclusion supporting the hypothesis!

Response 27: Dear reviewer, thank you for your suggestion. We have revised and supplemented the conclusion accordingly.

Round 2

Reviewer 2 Report

Authors replied in details and thoroughly in every comment, question, suggestion and objection of the review. Also they incorporated accordingly the suggested changed in the manuscript with clear and distinctive annotation. The resubmitted manuscript is totally different now, since all the interesting results, arose from laborious and high cost laboratory work, now can be supported by the relevant research on animals of the experiment.